# Obesity, the Adipose Organ and Cancer in Humans: Association or Causation?

**DOI:** 10.3390/biomedicines11051319

**Published:** 2023-04-28

**Authors:** Elisabetta Trevellin, Silvia Bettini, Anna Pilatone, Roberto Vettor, Gabriella Milan

**Affiliations:** Center for the Study and Integrated Treatment of Obesity (CeSTIO), Internal Medicine 3, Department of Medicine, University Hospital of Padova, 35128 Padova, Italy

**Keywords:** obesity, adipose organ, cancer

## Abstract

Epidemiological observations, experimental studies and clinical data show that obesity is associated with a higher risk of developing different types of cancer; however, proof of a cause–effect relationship that meets the causality criteria is still lacking. Several data suggest that the adipose organ could be the protagonist in this crosstalk. In particular, the adipose tissue (AT) alterations occurring in obesity parallel some tumour behaviours, such as their theoretically unlimited expandability, infiltration capacity, angiogenesis regulation, local and systemic inflammation and changes to the immunometabolism and secretome. Moreover, AT and cancer share similar morpho-functional units which regulate tissue expansion: the adiponiche and tumour-niche, respectively. Through direct and indirect interactions involving different cellular types and molecular mechanisms, the obesity-altered adiponiche contributes to cancer development, progression, metastasis and chemoresistance. Moreover, modifications to the gut microbiome and circadian rhythm disruption also play important roles. Clinical studies clearly demonstrate that weight loss is associated with a decreased risk of developing obesity-related cancers, matching the reverse-causality criteria and providing a causality correlation between the two variables. Here, we provide an overview of the methodological, epidemiological and pathophysiological aspects, with a special focus on clinical implications for cancer risk and prognosis and potential therapeutic interventions.

## 1. Causality Criteria in the Relationship between Obesity and Cancer

By itself, any statistical method cannot constitute proof that an association between two phenomena is based on a cause–effect relationship. A statistical association between two variables merely implies that knowing the value of one variable provides information about the value of the other, but it does not necessarily imply that one causes the other. In fact, the test must follow a method accepted in the scientific world; that is, verifying compliance with precise criteria of causality. Scientists have felt the need to establish rules to support cause–effect relationships, mostly in infectious diseases, with the Henle–Koch and Evans postulates [1] and the John Stuart Mill principles. The causality criteria, first elaborated by the English statistician Sir Austin Bradford Hill (1965) in a large study concerning the effect of smoking on humans [2], are now accepted by the scientific community and are based on frameworks used to assess whether an observed association is likely to be causal. Even if the most recent counterfactual approaches consider confounders and the collider bias (two possible issues that may lead to a non-causal association), a theory based on counterfactual outcomes is not the only approach available for reasoning about causes in epidemiology. A shared opinion is needed to adopt a pluralistic approach to causation and to collect different types of evidence in order to draw conclusions of a causal nature, particularly regarding complex exposures such as social context or climate change [3]. In this context, we face the challenge of obesity increasing throughout the world as an example of a complex variable that requires multilevel framing of the underlying causal processes, with structures extending from genetics to the environment.

According to the World Health Organization (WHO), overweight and obesity are defined as abnormal or excessive fat accumulation that may impair health [4]. The World Obesity Federation takes the position that obesity is a chronic, relapsing, progressive disease process [5]. Epidemiological studies have shown that obesity is associated with an increased risk of cancer, another chronic disease that is characterised by progression, relapse and a multifactorial origin. Although the mechanisms underlying this relationship are not fully understood, it is believed that in genetically predisposed individuals and in the presence of environmental permissive factors, obesity accelerates the development of cancer. The pathophysiological mechanisms underlying the development of several cancers in obesity are complex and incompletely understood. The expansion of the fat mass and its qualitative changes, collectively called adiposopathy [6], reflect a prototypical systemic metabolic and inflammatory disorder affecting many organs throughout the body. As described for the development of non-alcoholic fatty liver disease (NAFLD) and non-alcoholic steatohepatitis (NASH), the development of cancer in obesity is a result of multiple parallel hits [7] in which inflammation, fibrosis, the release of cytokines and adipokines, the gut microbiome, various dietary components, peripheral lipotoxicity, hormonal changes such as hyperinsulinemia, an increase in growth factors, the action of steroids and their tissue activity and several genetic and epigenetic modifications are critical drivers (Appendix A).

In order to establish a causal nature of the association between obesity and cancer, a process of causal inference must be performed, and the potential reversibility of the outcome due to the effect of therapeutic intervention on the exposure must be verified. While obesity may meet the criteria for a causal factor in the classic framework, if the interventions performed to achieve weight loss that are being compared are not explicitly specified (e.g., lifestyle, diet, hormones, genetic predisposition, etc.) there may be a violation of the consistency of counterfactual potential outcomes, which is a necessary condition for meaningful causal inference [8]. In fact, different methods of treating obesity may lead to different counterfactual mortality outcomes even if they lead to similar weight loss in each person. Therefore, the association between obesity and mortality offers little insight for preventive action. One alternative is to focus on the contrast between individuals who are randomly assigned to dietary modification versus those who are not, or the contrast between communities randomised to the taxation of sugary drinks versus those who are not taxed. The findings from such experiments would provide direct, actionable information about the effects of interventions against obesity. An observational study comparing people with and without obesity provides only indirect evidence and lacks a formally testable causal relation in the absence of further specification.

Given these premises, many resources were recently created to monitor the constantly growing amount of data regarding the association between potential risk factors and different types of cancer. One of the most important databases is the Continuous Update Project (CUP) of the American Institute for Cancer Research (AICR), which is a continually updated review of research that captures data from new scientific studies, randomised control trials and cohort studies from around the world as they are published for 17 different types of cancer [9]. Selecting “Body Fatness & Weight Gain” as risk factor, the key findings are that being overweight or obese throughout adulthood increases the risk of 14 different types of cancer: endometrial and ovarian cancer, oesophageal adenocarcinoma, gastric cancer (cardia), postmenopausal breast cancer, liver and colorectal cancer, kidney cancer, gallbladder, pancreatic and advanced prostate cancer, and mouth, pharynx and larynx cancer. In the present manuscript, we also considered thyroid cancer, multiple myeloma and lung cancer, which were not included in the CUP analysis but are relevant to understand the conflicting relationship between cancer and obesity (Figure 1). 

The association between overweight/obesity and cancer risk, however, is not always consistent: for example, overweight and obesity prior to menopause decreases the risk of premenopausal breast cancer, and during the period of 18–30 years of age, it decreases the risk of pre- and postmenopausal breast cancer [10]. Furthermore, increasing evidence has been observed in different studies that the overweight condition may be a protective factor for overall survival after neoadjuvant therapy and surgical resection in oesophageal cancer [11]; after the immunotherapy treatment of renal cell carcinoma (RCC) [12]; for overall survival, disease-free survival and cancer-specific survival in colorectal cancer (CRC) [13] and for a better prognosis in other types of cancer [14,15,16,17,18,19]. A systematic review and a meta-analysis of 203 studies considering 6.3 million cancer patients showed that obesity was associated with higher overall mortality in breast, CRC and endometrial cancer [20]. The data were compared with patients without obesity. Furthermore, studies indicate that obesity may worsen several aspects for cancer survivors, including their quality of life, cancer recurrence, cancer progression, prognosis and risk of second primary cancers [20,21]. On the other hand, patients with obesity and lung cancer, RCC or melanoma had better survival outcomes when compared with patients without obesity and with the same cancer [20].

The fact that obesity is associated with a more favourable prognosis despite being a risk factor for many types of cancer is a well-known phenomenon called “the obesity paradox”, and it highlights the need to evaluate the clinical relevance of host factors, such as body size and composition or adipose tissue quality, and their potential contribution to tumour biology, prognosis and treatment selection [22]. This paradox may be explained by the limitations of Body Mass Index (BMI), a simplistic index defined as a person’s weight in kilograms divided by the square of their height in metres, which is commonly used to classify overweight (>25 kg/m^2^) and obesity (>30 kg/m^2^) in adults. Indeed, BMI does not take into account age, hormonal profile, inflammatory status or adipose tissue distribution and quality [23]; the latter is a more reliable factor that has been independently associated with insulin resistance and dyslipidemia and has been proven to be the main determinant of most pathophysiological mechanisms linking obesity to obesity-related diseases, including cancer [24]. Moreover, other confounding variables (e.g., smoking; alcohol consumption; other comorbidities), reverse causation (weight change as a consequence rather than a cause of cancer) and collider biases may be reasons why the paradox is observed in different studies, although they may only partially explain how the direction of a true association can be reversed and may not fully account for it [25].

## 2. Obesity and Cancer Risk: Epidemiology and Controversial Evidence

The increase in the risk of cancer in obesity has been well-established for many years [26], and in an American study, 14% of cancer deaths in men and 20% in women were attributed to obesity [27]. An association between BMI and an increased incidence of cancer has been demonstrated [28,29], and a meta-analysis by the International Agency for Research on Cancer (IARC) working group concluded that the absence of excess body fatness lowers the risk of most cancers [30]. Obesity is characterised by a state of low-grade inflammation that is a common factor for many chronic diseases, including cancer. Furthermore, the role of inflammatory and metabolic biomarkers goes beyond the crude measure of BMI as a predictor of cancer mortality risk [31]. In fact, recent studies have suggested that normal-weight individuals with metabolic dysregulation and metabolically unhealthy patients with obesity presented an increased risk of cancer [32,33,34]. On the other hand, it has been well-established that obesity may guarantee a better performance status in patients with cancer and could improve their prognosis, as previously mentioned. Moreover, recent studies based on Mendelian randomization (MR), a method that employs genetic variants to remove biases due to confounding and reverse causation to estimate the direct and indirect effects of the exposure on an outcome [35,36], suggested that in contrast to the positive association in conventional observational studies, BMI may even show inverse associations with some types of cancer (e.g., early life obesity with postmenopausal breast cancer [37]). In addition, MR analysis further established the causality of obesity with colorectal cancer, endometrial cancer, ovarian cancer, oesophageal adenocarcinoma, kidney cancer and pancreatic cancer, but highlighted obesity’s inverse relationship with other types of cancer that are not related to obesity. Furthermore, MR studies did not confirm a positive association between obesity and gallbladder cancer, gastric cardia cancer and multiple myeloma [38]. In this review, we aim to consider 17 different histotypes of cancers in order to discuss this ambivalence (Figure 1 and Table 1).

## 3. The Adipose Organ and Cancer: General Aspects and Specific Mechanisms

The adipose organ is composed of several AT depots with a specific anatomical localisation, distinct expression and secretion profiles and peculiar metabolic and endocrine functions [77]. Particularly, the adipose organ consists of white, brown and beige adipocytes, which play different roles in energy storage, consumption and thermogenesis, contributing to whole-body metabolism [78]. Several studies highlighted the presence of other cell types embedded in the AT, such as adipogenic precursors and mature endothelial, immune and nervous system cells, which have significant relevance in regulating adipose organ growth [79]. Our group recently described the AT stem cell niche (called the “adiponiche”) in the context of obesity and metabolic diseases. The term niche refers to the anatomical and functional unit that regulates stem cell homeostasis and behaviour. We recapitulated evidence supporting the hypothesis that adipose stromal/stem cells (ASCs) are regulated by both cell-to-cell interactions and their microenvironment in order to support physiological AT growth and to maintain an energy balance in relation to the environmental conditions, such as the availability of nutrients, temperature changes and physical activity. Each element of the adiponiche can become dysfunctional, contributing to the development of obesity and metabolic complications correlated with many pathological alterations described in the AT of patients with obesity [80]. A number of direct and indirect mechanisms have been proposed to explain the role of obesity in cancer development, progression, invasiveness and chemoresistance. Many studies focused on the crosstalk between peritumoral AT and cancer cells, while others investigated the systemic impact of fat mass expansion, particularly with respect to circulatory factors (e.g, adipokines, cytokines, lipids, etc.) and hyperinsulinemia, leading to the creation of a cancer-permissive milieu [81,82,83,84,85]. Interestingly, AT modifications occurring in obesity display some similarities with the behaviour of solid tumours [86]. These similarities include: the theoretically unlimited expandability, the capacity to infiltrate other tissues and to induce angiogenesis, the local and systemic inflammatory process and changes in the immune system, the induction of DNA damage and resistance to apoptosis and the ability to influence the function of several distant organs due to the secretion of hormones and miRNAs. The adipose organ and cancer also share some elementary morphofunctional tissue aspects, such as the presence of a tissue niche. In the present review, we centre on the main general processes and specific molecular mechanisms involving cells of the obesity-altered adiponiche in the context of so-called “adiposopathy” [87,88], underlining innovative issues and the results of our recent research (Figure 2).

### 3.1. Hypoxia

Obesity is characterised by rapid AT expansion. However, the capillary networks fail to match their growth with the rapid expansion of AT depots and are instead reported to undergo rarefaction upon obesity [89,90]. This effect results in regions in which the vascular supply is insufficient, producing areas of hypoxia that mimic the ones present in growing tumours (Figure 2). For this reason, defining how obesity affects vascular function is relevant to understanding the adipose organ–cancer relationship. In both obesity and cancer, hypoxia triggers the induction of pro-angiogenic factors such as vascular endothelial growth factor (VEGF) and hypoxia-inducible factor 1α (HIF-1α). For example, HIF-1 plays key roles in many crucial aspects of breast cancer biology, including angiogenesis, stem cell maintenance, metabolic reprogramming, the epithelial-to-mesenchymal transition (EMT), invasion, metastasis and resistance to radiation therapy and chemotherapy [91]. However, the angiogenic stimulus is unable to sufficiently compensate and reverse AT hypoxia (the blood vessels function poorly; thus, low oxygen levels persist), leading to a chronic condition that contributes to adipocyte death and eventually supports the infiltration of macrophages [87].

### 3.2. Inflammation, Fibrosis and Senescence

The previously described hypoxia induces the dysfunctional AT to release pro-inflammatory cytokines, which promote the development of low-grade chronic inflammation and enhance fibrosis, two key processes involved in carcinogenesis [92]. In particular, tumour necrosis factor α (TNF-α) leads to the activation of the nuclear factor (NF)-κB, with the subsequent upregulation of several negative regulators of apoptosis, promoting cell survival and cancer progression; interleukin 6 (IL-6) activates the Janus kinase (JAK)/signal transducer and the activator of transcription 3 (STAT3) pathway, leading to cell survival and proliferation [93]. This obesity-induced inflammation disrupts the immunometabolism, the bidirectional interaction between the immune system and the whole-body metabolism [94,95]. This could contribute to the development of cancer [96]. Other components include matrix metalloproteinases (MMPs), which are associated with adipocyte differentiation, cancer cell invasion and metastasis, immune cells and pro-inflammatory M1-polarised AT macrophages (ATMs), reduced levels of regulatory T cells (Tregs) and M2-polarised anti-inflammatory ATMs (Figure 2). 

In the lean state, AT T helper 2 (TH2) cells, Tregs, eosinophils and M2-like resident macrophages predominate, with Tregs that secrete interleukin 10 (IL-10) stimulating, in turn, the secretion of IL-10 from resident M2-like macrophages. Eosinophils secrete interleukin 4 (IL-4) and interleukin 13 (IL-13), further contributing to the anti-inflammatory, insulin-sensitive phenotype. In obesity-induced inflammation, monocytes respond to chemotactic signals, transmigrate into the adipose tissues and become polarised into the highly pro-inflammatory, M1-like state. Once recruited, these M1-like ATMs secrete pro-inflammatory cytokines that work in a paracrine fashion. The eosinophil content declines, and there is a shift in adipose tissue T cell populations, with a decrease in the content of Tregs and an increase in CD4+ T helper 1 (TH1) and CD8+ effector T cells, which secrete pro-inflammatory cytokines. B cell numbers also increase and activate T cells, which potentiate M1-like macrophage polarisation, inflammation and insulin resistance [97]. Specifically, M1-polarised ATMs contribute to extracellular matrix (ECM) remodelling in obese tissues due to the accumulation of collagen type 6 (COL6α3), which is a fibrotic ECM protein involved in AT stiffness and metabolic dysfunction. A COL6α3 cleavage product, endothrophin, worsens fibrosis and inflammation, thus contributing to changes in the activities of M1-polarized ATMs. In particular, the latter begin to display a blunted phagocytotic capacity and develop a senescence-associated secretory phenotype (SASP) [98] that acts in a paracrine and autocrine manner to spread and strengthen senescence-inducing signals to neighbouring adipose tissue cells [99,100,101,102]. These macrophages surround dying and dead adipocytes, organising pathologically in crown-like structures (CLSs) which are involved, again, in the overproduction of pro-inflammatory cytokines [103]. This state of unresolved chronic inflammation, with an accumulation of SAPS and the formation of CLSs, is considered a central driver of pathological tissue remodelling, the development of adipose tissue fibrosis, metabolic dysfunction and cancer [104]. 

In this context, inflammasomes (the cytosolic multiprotein oligomeric components of the innate immune system), play an important role in cancer development, showing tumour-promoting actions [105]. The inflammasomes, which are activated by fatty acids and high glucose levels, induce the secretion of the inflammatory cytokines IL-1β and IL-18 and control adipocyte differentiation and insulin sensitivity [106], leading to the infiltration of more immune cells and resulting in the generation and maintenance of an inflammatory microenvironment surrounding the cancer cells. It has been suggested that inflammasomes promote proliferation and the invasion of colon cancer cells through the activation of IL-1β [107], emphasising their potential crucial role in the link between obesity, inflammation and carcinogenesis. This disruption of the homeostatic balance between pro- and anti-inflammatory cellular mediators in adipose tissue is a basis for the development of metabolic disorders and may have important implications for cancer metastasis [108].

### 3.3. ROS Production and Mitochondrial Alterations

The inflammation-related mechanisms that occur in obesity may increase the production of ROS and, in turn, excess ROS may lead to mitochondrial dysfunction (Figure 2). The production of ROS in the mitochondria depends on the physiological or pathological conditions of the cells. In particular, in adipose precursor cells, the redox-sensitive signalling molecules phosphatidylinositol-3-kinase (PI3K) and mitogen-activated protein kinases (MAPK) take part in the insulin-signalling pathway, whereas peroxisome proliferator-activated receptor γ (PPARγ) and CCAAT/enhancer binding protein β (C/EBPβ) are fundamental regulators of adipogenesis. Obesity-related oxidative stress is associated with an elevated expression of these molecules in pre-adipocytes, leading to the altered signal transduction and dysregulation of adipocyte differentiation. Moreover, when the generation of ROS increases, several constituents of the respiratory chain and Krebs cycle enzymes may lose activity, thus leading to mitochondrial dysfunction linked to altered adipogenesis, glucose and lipid metabolisms, adiponectin production and an elevated rate of apoptosis. These alterations, associated with mitochondrial dysfunction and elevated levels of ROS, also induce the endoplasmic reticulum (ER) stress that results in modifications of the activity of the ER calcium ATPase and the subsequent disequilibrium in calcium homeostasis [109]. These conditions, as part of oxidative stress and chronic inflammation, may also create a microenvironment favourable to tumour development in obesity [110].

### 3.4. Altered Secretion of Adipocyte-Derived Factors

Adipocytes are the most important components of adipose tissue and are also the conductors of microenvironmental evolution during obesity. These cells can affect tumour growth at multiple levels through paracrine, autocrine and endocrine factors such as adipokines, extracellular matrix (ECM) components, hormonal factors (e.g., via the action of aromatase, resulting in increased oestrogen levels in the microenvironment), growth factors and inflammatory cytokines, and the direct supply of rate-limiting metabolites released by the adipocytes, such as free fatty acids (FFAs) [111]. In particular, the most important role in tumorigenesis is played by adipokines, which can affect tumour biology by regulating insulin resistance and inflammation. For example, adiponectin, which promotes adipose tissue microenvironment homeostasis, acts locally on adipocytes to regulate glucose uptake, adipogenesis and lipid storage, whereas leptin, which promotes adipose tissue microenvironment dysfunction, regulates adipocyte lipolysis, producing adipocyte-derived lipids that are transferred to tumour cells to induce metabolic reprogramming, growth and invasion [87]. Both these adipokines play roles in the tumorigenesis of different cancer types. In postmenopausal breast cancer, increased leptin synthesis by breast AT contributes to the upregulation of pro-inflammatory cytokines as well as cyclooxygenase-2 (COX-2) and prostaglandin E2 (PGE2). These factors potentiate the expression of aromatase in breast adipose stromal cells, a central enzyme in the production of oestrogens, particularly oestradiol, which is increased as a result [112]. Considering the essential role of these hormones in the normal development of the mammary gland, several studies associated cytokine-induced aromatase upregulation with breast cancer growth and progression [113]. This relationship is supported by the successful results obtained after treating patients with aromatase inhibitors (Ais). Indeed, most clinical studies proved that blocking aromatase activity greatly reduces the amount of oestrogen in the body, reducing breast cancer recurrence compared with standard Tamoxifen therapy [114]. In animal studies, the loss of adiponectin enhances the development of colitis-associated colorectal cancer [115], and diet-induced obesity elevates colonic TNF-α in mice [116]. TNF-α and IL-6 are induced by leptin, which stimulated the formation of lipid droplets and increased cell proliferation in intestinal epithelial cells in an in vitro study [117]. Leptin and adiponectin have also been linked to the presence of Barrett’s oesophagus and its progression to EAC [118]. In the development of RCC, data suggest that while the adiponectin released from perinephric adipose tissue may impact RCC aggressiveness via the alteration of the tumour microenvironment, the levels of adiponectin in the perinephric-fat-conditioned medium appear to not be considerably related to the aggressiveness of RCC [119]. In multiple myeloma, adipocyte-secreted Angiotensin II (ANG-II) directly stimulates acetyl-CoA synthetase 2 (ACSS2) in myeloma cells, contributing to obesity-induced myelomatous tumorigenesis, and the inhibition of the ANG-II-ACSS2 axis prevents obesity-induced tumour growth [120].

Our previously published data showed that in patients affected by EAC, peritumoral AT displays an increased adipocyte size, augmented production of leptin, an increased expression of angiogenesis, lymphangiogenesis and inflammation markers, and the upregulation of genes involved in stemness and EMT. This dysfunctional AT was associated with lymph node invasion [121] and chemoresistance [122] and was not observed in a fat depot distant from the tumour of the same patient. In particular, during the carcinogenesis process, peritumoral AT progressively increases the expression of leptin, CD31, VEGF, VEGF-C, CD45, CD34, nucleostemin (NSTM), octamer-binding transcription factor 4 (OCT-4) and alpha-smooth muscle actin (α-SMA) under the stimulation of tumour-derived factors or AT-secreted adipokines such as leptin. In the presence of neoadjuvant therapy, some of these effects decrease in peritumoral AT; however, in patients with a poor response to treatment, the expression of factors involved in stemness, EMT, neo-angiogenesis and leptin signalling remains higher in comparison with patients with a better prognosis, suggesting that peritumoral adipose tissue may promote characteristics of the microenvironment that sustain the progression and chemoresistance of the tumour [122]. These data indicate that contiguous visceral adipose tissue may directly affect tumour invasiveness through the action of leptin, and recent additional results support the hypothesis that adipose-specific secreted metabolites influence cancer progression and metabolism in a depot-specific manner (personal, unpublished data) (Figure 3). 

### 3.5. The Protein Kinase CK2

As previously described, adipocyte-derived lipids play a role in cancer development through the involvement of pro-survival protein kinases that contribute to the regulation of aberrant lipid metabolism in malignant cells. In this context, casein kinase 2 (CK2) plays a pivotal role, regulating intracellular processes linked to lipid accumulation and mobilisation in normal tissues as well as in disease states, particularly cancer [123]. This Ser/Thr protein kinase acts in a wide spectrum of signal transduction pathways whose dysregulation leads to malignancy, such as anti-apoptotic signals, cell survival, growth and proliferation, angiogenesis, DNA repair, cell migration and invasiveness [124]. Multiple studies demonstrated that CK2 is usually overexpressed in cancer, leading to the upregulation of oncogenic molecules such as protein-kinase B (AKT), Wnt and extracellular signal-regulated kinase (ERK) and to the downregulation of tumour suppressor molecules (such as Notch and Ikaros) [125]. The degree to which CK2 potentiates each signal depends on its expression/activity level, and its targeting may represent a general strategy for treating neoplasia in different contexts. Our group and another recently demonstrated that CK2 also plays relevant roles in the adipose organ, as it is upregulated in the AT of patients with obesity, supporting both adipocyte hypertrophy/hyperplasia and adipogenesis, which are essential for the AT expansion [123,126]. It was demonstrated that CK2 is preferentially activated in white adipocytes [127], is crucial for adipogenic differentiation [128], positively modulates the insulin-stimulated glucose uptake of adipocytes [126] and is hyperactivated in the lipomatous tissue of patients with multiple symmetric lipomatosis, a rare disorder characterised by the growth of non-encapsulated masses of subcutaneous AT [129]. In addition, in the context of adipose tissue undergoing excessive lipid accumulation and chronic low-grade inflammation, there is evidence of crosstalk between CK2 and adipokines (leptin and adiponectin) and between CK2 and cytokines (as interferon-γ, IFN-γ and IL-6) [123]. Lastly, CK2 is involved in the ASC signalling pathway mediated by adiponectin, contributing to adiponiche regulation [130] (Figure 3).

### 3.6. The Adipose Tissue Stromal Stem Cells

In addition to the pro-tumorigenic paracrine and/or endocrine effects of inflammatory factors from dysfunctional adipose tissue, several studies have shown that ASCs actually infiltrate cancer lesions and actively contribute to a cancer-promoting microenvironment via paracrine and contact-dependent effects [85]. We observed an increased number of ASCs and a decreased number of capillaries in patients with obesity compared to lean subjects in both subcutaneous and in visceral AT [90]. Similar results were described in animal models of obesity, and an increased frequency of ASCs was detected in the circulation of patients with obesity and colorectal cancer [131,132]. These data suggest that an increased number of AT-resident and circulating ASCs altered in their biological features by the obesity environment could affect tumour growth, acting on cancer cell proliferation and angiogenesis within the tumour, metastasis and drug resistance [133]. In particular, several in vitro studies showed that ASCs isolated from the AT of patients with obesity displayed an altered gene expression profile resulting in the production of high levels of tumorigenic factors, including leptin, inflammatory cytokines, adhesion molecules, growth factors, ECM components and remodelling proteins [133,134]. Obesity-altered ASCs have an increased potential to spread cancer cells and become an integral constituent of the tumour microenvironment (TME). It is important to note that ASCs can be mobilised and recruited to the tumour from both local and distal AT depots. In this context, ASCs play a central role in tumorigenesis via their secretome and by interacting with tumour cells and immune system cells in various cancer types, such as postmenopausal breast, colorectal, prostate, ovarian, multiple myeloma, cervical, bladder and gastrointestinal cancers [133]. Interestingly, a parallel exists between matrix remodelling in obesity and cancer. Many ECM proteins upregulated in AT during obesity are linked with inflammation, the recruiting and conditioning of immune cells and with the promotion of cancer stemness [135,136,137]. ASCs produce a large amount of ECM molecules which can alter TME stiffness, promoting myofibroblast differentiation via mechano-trasduction [133]. Furthermore, the tumour represents a non-healing wound in which immune cells directly influence tumorigenicity and cancer development. In this context, the immunomodulatory properties of ASCs could be integral to cancer progression [138]. Additionally, ASCs enhance the migration of cancer cells, increasing the number of metastases, and influence angiogenesis in different tumour models [133]. Lastly, in preclinical models, obesity also increases the capacity of ASCs to differentiate into cancer/carcinoma-associated fibroblasts (CAFs), which are present in more invasive tumours, enhancing cancer cell proliferation, invasion and metastasis. The origin of CAFs is heterogeneous, varies between different tumour histotypes and is critical in determining the degree of CAF malignancy [139]. Recent studies have shown that mature adipocytes, particularly obesity-altered adipocytes, can also de-differentiate and acquire a myofibroblast/fibroblast phenotype [137]. Furthermore, cancer cells themselves “educate” ASCs through secreted factors and exosomes carrying specific miRNA and lncRNA, promoting their differentiation to CAF, reprogramming their cellular metabolism and influencing their migratory capability [139]. 

All these observations suggest that by acting through similar mechanisms, obesity-altered ASCs could be partially responsible for the dysfunction of both the AT niche and tumour niche in patients with obesity and cancer. This point must be carefully taken into account to evaluate the effectiveness and safety of using of ASCs for the treatment of diseases, particularly cancers, and in regenerative medicine [140].

## 4. The Involvement of Microbiota Dysregulation

The human gut microbiota is a complex ecosystem composed of approximately 1014 microorganisms (bacteria, archaea and eukarya) that colonise the gastrointestinal (GI) tract and produce metabolites, such as short-chain fatty acids (SCFAs), which are crucial for GI health. Changes in the composition of the gut microbiota (gut dysbiosis) affect the human body’s energy and immune homeostasis, playing an important role in the development of obesity and related diseases, such as cancer [141,142]. The role of gut dysbiosis in obesity has been demonstrated well in mice, and it has been observed that the transplantation of wild-type or obese microbiota into germ-free mice results in a normalised or increased body weight, respectively, in the transplanted animals. Moreover, changes in the abundance of different microbial phyla have been reported in mouse models after the administration of a high-fat diet (HFD), suggesting that dietary patterns can also change the microbial composition, further augmenting a propensity towards excess bodyweight [143]. Specifically, in obesity, the SCFAs produced by bacteria are decreased, thus reducing their previously described protective role based on G protein signalling [144]. Indeed, the effects of SCFA supplementation on the expression of the G protein receptor (GPR) and the gut microbiota composition may further result in body weight reduction by enhancing triglyceride hydrolysis and FFA oxidation in the adipose tissue, promoting beige adipogenesis and mitochondrial biogenesis and inhibiting chronic inflammation [145]. 

Several studies associated gastrointestinal cancers (such as colon, liver and pancreatic cancers) with intestinal microbiota and dysbiosis [31,146,147]. In particular, microbiome-derived peptides modulate immune cell activity, and the leakage of lipopolysaccharides (LPSs) activates TLR4 and nuclear factor-kB (NF-kB), leading to inflammatory-induced carcinogenesis in the colon and the liver [147,148]. For example, in the development of HCC, obesity-induced insulin resistance amplifies de novo lipogenesis and increases FFA flux, which also derives from an altered gut microbiome, leading to mitochondrial dysfunction with oxidative stress, ER stress and activating the unfolded protein response, all resulting in hepatic inflammation. In addition, small bowel permeability can be enhanced through the consequent raised circulating levels of molecules, which contribute to the activation of the inflammasome. Furthermore, bacterial products such as LPSs induce the release of cytokines from Kupffer cells [149]. These factors all contribute to a cycle of hepatocyte damage and regeneration which create a pro-tumorigenic microenvironment [150]. 

## 5. Circadian Rhythms in Obesity and Cancer Progression

We wanted to explore the role of circadian rhythm imbalance, which has been identified as a risk factor for the development of obesity (for example, in shift workers). Circadian clocks are cell-autonomous timing systems that generate an approximately 24 h periodic rhythm, i.e., the circadian rhythm. Disturbances of this rhythm caused by sleep deprivation or eating at night are closely associated with the development of sleep and mood disorders, obesity, diabetes and cancers [151]. Recent findings provide evidence that exposure to artificial light at night, which is emitted by residential areas, road illumination and non-stop economic activities, represents a remarkable risk factor for weight gain, overweight and obesity and cancer [152]. Many studies in humans have shown that obesity associates with disturbances in biological rhythms, such as those governing sleep and food intake. It has been shown that people with obesity have delayed bedtimes and altered sleep duration compared to healthy people, due to the disruption of the plasma melatonin circadian rhythm and night eating syndrome, which is characterised by nocturnal hyperphagia and morning anorexia. In brief, obesity is involved not only in the malfunction of the metabolic system but also in the disturbance of biological rhythms [153]. Furthermore, several studies showed that disrupted circadian rhythmicity can impact the pathogenesis of cancer due to the dysregulation of the crosstalk between the circadian clock machinery and the cell cycle, DNA repair, apoptosis, senescence, autophagy and other oncogenic and immune pathways. These alterations lead to uncontrolled proliferation, escape from apoptosis, metastatic spread, immune evasion, enhanced angiogenesis and anticancer drug resistance, which are all hallmarks of cancer [151]. Although there are few data regarding the impact of the circadian rhythm on obesity-associated tumours, recent studies have reported that circadian disruption affects breast cancer development and the spread of metastases, which occur more commonly during the sleep cycle [154]. Interestingly, it was demonstrated that alterations to biological rhythms promoted tissue metastasis, especially distal lymph node metastasis, further causing a decrease in survival [155]. Particularly, in both patients and mouse models, circulating tumour cells (CTCs) disseminated during sleep displayed a greater metastatic ability compared to those circulating during the active-phase due to a marked upregulation of mitotic genes exclusively during the rest phase. Moreover, it was shown that key circadian rhythm hormones (melatonin, testosterone and glucocorticoids) dictate the generation dynamics of CTCs, and insulin directly promotes tumour cell proliferation in a time-dependent manner. These data suggest the importance of considering timing for both biological sample collection and for the administration of therapy in oncological patients [154].

## 6. Effects of Adipose Tissue on Cancer Cachexia

Cancer-associated cachexia is a complex syndrome, involving multiple organs and tissues, that is characterised by the specific loss of AT and skeletal muscle mass and is associated with decreased survival and a poor response to chemotherapy [156]. Recent findings showed that white, brown and intermediate beige AT are deeply involved in a dynamic reprogramming of the energy metabolism during cachexia [157]. Both the macroenvironment and microenvironment contribute to the inflammatory process that characterizes cancer-associated cachexia [158], and the adipose organ plays a critical role by increasing lipid catabolism and mobilization concomitantly with a decreased lipid uptake through lipoprotein lipase and reduced rates of lipogenesis [159]. This dramatic decrease in white adipose depots is combined with adipocyte browning and the stimulation of UCP1, which, in turn, increases thermogenesis and high energy expenditure [160]. 

Moreover, in cancer cachexia, the release of pro-inflammatory cytokines and morphological and structural modifications resulting in ECM rearrangement and fibrosis have been demonstrated in AT [161]. These conditions parallel a status of muscle mass dysfunction and atrophy, with a concomitant infiltration of lipid droplets derived from AT lipolysis, that further impairs muscle tissue structure and strength [162]. These alterations have also been observed in sarcopenic obesity [163], and although the role of AT in cancer-associated cachexia has been thoroughly discussed [164], its role is still controversial. On one hand, it seems that metabolic dysfunction, inflammation and insulin resistance (all hallmarks of obesity) accelerate the wasting of fat and lean mass and worsen the prognosis of patients [165,166,167], thus suggesting that obesity therapy could also counteract the onset of cancer-associated cachexia. Conversely, it has been suggested that the remodelling of AT, in terms of an increased macrophage abundance (stimulated by hypoxia [168]) or in terms of the attenuation of browning (stimulated by immune-metabolic dysfunction [169]), may play a protective role against cancer-associated cachexia and increase patient survival.

## 7. Effects of Obesity Treatment on Cancer Incidence and Prevention

Since obesity represents a potential risk factor for the development of cancer, strategies that aim to treat obesity could potentially be useful for limiting the spread of cancer. If the reversal of exposure (obesity) decreases the incidence of outcome (cancer), it can be speculated that causation underlies their relationship (see Figure 1). A meta-analysis reported a sufficient strength of evidence for a cancer-preventive effect of the avoidance of weight gain on numerous types of cancer [30]. Lifestyle modifications (diet and physical activity), pharmacotherapy and bariatric surgery are the three main therapeutic interventions for treating overweight and obesity, thus reducing the risk of cardiovascular disease (CVD), the incidence of type 2 diabetes (T2DM) and the risk of all-cause mortality [170,171,172]. Obesity may have a causal and reversible effect on cancer risk; however, when comparing the cancer-related mortality after weight loss, the results are controversial. In this regard, some observational studies have provided consistent evidence that individuals who have lost weight have a lower risk of breast [173] and endometrial [174] cancer compared to patients who did not lose weight. Regarding prostate cancer, men who maintained more than 11 pounds of weight loss over a 10-year period had a decreased risk of non-metastatic, high-grade prostate cancer [175]. However, while a high BMI was positively associated with non-metastatic, high-grade prostate cancer, it was simultaneously inversely associated with the risk of developing low-grade disease. The “Look AHEAD”, a large, randomised trial on the effects of lifestyle intervention on cancer outcomes, assigned patients with overweight/obesity and T2DM to an intensive lifestyle intervention (ILI), targeting a weight loss of 7%, or to “support and education” for T2DM (DSE) [176]. Obesity-related incident cancer was similar between the two subgroups (ILI, *n* = 158 vs. DSE, *n* = 185, hazard ratio, 0.84; 95% CI: 0.68–1.04; *p* = 0.10). Indeed, cancer-specific mortality was ILI, *n* = 80 vs. DSE, *n* = 85 (hazard ratio, 0.92; 95% CI: 0.68–1.25; *p* = 0.59). A limitation of the Look AHEAD trial was the presence of only patients with T2DM. It would be a challenge to perform future trials investigating the effects of weight loss on cancer risk by specific cancer site and subtype, in patients with obesity but without T2DM and while correcting the data for sex, age, race and ethnicity. 

While diet or certain unhealthy foods have been shown to be more associated with a higher cancer risk [177], weight loss through dietary interventions such as caloric restriction, a ketogenic diet or time-restricted eating (a type of intermittent fasting, which involves consuming all calories within a consistent 8–12 h daily window based on the normal circadian rhythm of eating) showed beneficial metabolic effects and decreased the risk of cancer [178,179].

Currently, bariatric surgery is the most effective treatment for obesity, guaranteeing weight loss of up to approximately 35% of the body weight, with maintenance over many years [172,180]. The short- and long-term effectiveness of bariatric surgery are due to both the caloric restriction and the modification of different metabolic regulators acting at the GI and central levels [181]. Among these mechanisms, it has been demonstrated that bariatric surgery induces important changes in the species of gut microbiota [31]. Despite the fact that some large studies have proven the association between bariatric-surgery-induced weight loss and a reduced incidence of obesity-related cancer [182,183,184], few studies are available regarding the effect of weight loss on cancer-related mortality [185,186], and some results are discordant. For example, retrospective studies have suggested that bariatric surgery is significantly associated with a decreased incidence of cancer in women but not in men [180,187], or in men but not in women [188] or not significantly associated, as observed in CRC [189]. Conversely, a prospective study demonstrated a time-dependent increase in the risk of CRC 10 years or more after bariatric surgery [190]. According to some authors, the beneficial effects of bariatric surgery were strongest for female obesity-related tumours compared to males, especially in the postmenopausal subgroup. This positive outcome in females is likely to be associated with a substantial decrease in the production of sex hormones and bioavailability, inducing a decrease in hormone-sensitive breast and endometrial cancers after surgery [191]. A meta-analysis showed a 60% reduction in the risk of developing endometrial cancer among patients undergoing bariatric surgery compared with women who had not [192]. An observational study comparing patients who received surgery to a control group who did not displayed that the cancer incidence declined only for obesity-related malignancies (i.e., postmenopausal breast, oesophagus, colorectal, liver and pancreas), even if mortality dropped for all cancer types [193].

A recent retrospective, observational, matched cohort study in adult patients with obesity who underwent bariatric surgery or received typical care (Surgical Procedures and Long-term Effectiveness in Neoplastic Disease Incidence and Death, SPLENDID) showed a significantly lower incidence of obesity-associated cancer and cancer-related mortality in the bariatric surgery subgroup [194]. Despite the entity of weight loss being associated with a reduced obesity-related cancer mortality in a dose-dependent manner, the analysis had several limitations and, in the absence of randomised studies (despite corrections in statistical models and sensitivity), it is not possible to assess whether the patients undergoing bariatric surgery have different characteristics (e.g., social, economic and cultural conditions) in comparison with the control group or whether the lifestyle changes that surgical candidates were required to make could independently contribute to risk reduction. 

## 8. Concluding Remarks: Is There Evidence of a Cause–Effect Relationship?

Epidemiological data strongly associate overweight and obesity with an increased risk of developing several types of cancer. Experimental data indicate that the adipose organ is the key player in the relationship between obesity and cancer in humans. Among several cellular processes and molecular mechanisms summarized in the present review, we would like to underline the activation of the cellular senescence program recently observed in different cell types (progenitors and mature adipocyte and immune cells) of the obesity-altered adiponiche. Moreover, our group and others demonstrated a central role of CK2 in AT, obesity and adipogenesis, and we provided additional insights on the role of leptin, which results in increased peritumoral AT, in cancer progression and chemo-resistance. Lastly, considering the experimental data and clinical evaluations, we suggested that systemic alterations due to the dysregulation of the microbiota and circadian rhythm could play a relevant role in both obesity and in cancer progression. Clinical data suggest that weight loss itself, especially when it results in a substantial visceral fat mass loss rather than procedure-specific mechanisms, is the main determinant of cancer risk reduction in patients with obesity. However, if the basis of this association is an undisputable cause–effect relationship, the intervention of reverting the features of obesity alone should be sufficient to reduce the incidence of cancer in patients recovering from obesity to the value observed in the normal-weight population, thus demonstrating the reversibility of the process (see Figure 1). However, this is not always the case, suggesting that the effects of obesity on cancer risk may differ by molecular tumour subtype, sex, age, the presence of comorbidities and other factors.

In this context, many of the studies considered in the present manuscript did not perform a comprehensive clinical evaluation of the obesity disease by accurately assessing body fatness and composition, staging and gender specificity and taking into account metabolic and cardiovascular alterations. The limitation of the use of the BMI only to classify patients with obesity appears to be increasingly evident and must be overcome. For example, rather than simply categorising patients based on anthropometric measures, the Edmonton Obesity Staging System (EOSS) provides a five-stage classification of obesity via assessing the presence and severity of risk factors, comorbidities and functional and psychological limitations [195]. The EOSS is based on the evaluation of medical history, clinical and functional measurements and simple, routine diagnostic investigations that are easily and widely available. However, it still lacks a strong independent predictor of all-cause mortality due to the flexibility in determining functional impairment. Recently, we suggested an improvement of this staging system that considers cardiorespiratory fitness (CRF), which is measured by exercise testing patients [196]. Nevertheless, the measurement of obesity and metabolic impairment is still primarily based on BMI classes, and a great deal of effort is urgently needed from clinicians to address this shortcoming.

In the meantime, recent computational approaches have been developed in the emerging field of digital pathology in order to explain the cell-to-cell interaction within the TME [197]: for example, an agent-based model of the dynamics involved in the immune response has proved to be useful for the in silico study of the tumour behaviour in the context of the obesity-induced alteration of the immune response in peritumoral AT [198]. These methods are powerful tools for capturing a deeper degree of heterogeneity and specific characteristics of the tumour microenvironment that are unique for each patient and depend on their genome, microbiome, disease history, lifestyle and environment. These characteristics may be difficult to retrieve in experiments and clinical trials, especially the temporal dynamics of spatial heterogeneity; therefore, these agent-based computational models are suited to tackling this challenge.

Future randomised studies should be designed to test the effects of weight loss by specific cancer site and molecular subtype, in patients matched for sex, age, race and ethnicity and in specific subgroups of patients with different degrees of metabolic complications to determine whether weight loss provides similar benefits across individual cancers and population subcategories [21]. Furthermore, clinical data should be integrated, whenever possible, with a parallel molecular and cellular analysis of the adipose organ of each patient and compared with substantial metabolic parameters. A more accurate integration of such findings by scientists and clinicians, combined with the assessment of clinically valuable ranges and cut-offs of reliable measurements of obesity, will be the only method for assessing whether obesity has a causal—and plausibly reversible—effect on cancer development.

## Figures and Tables

**Figure 1 biomedicines-11-01319-f001:**
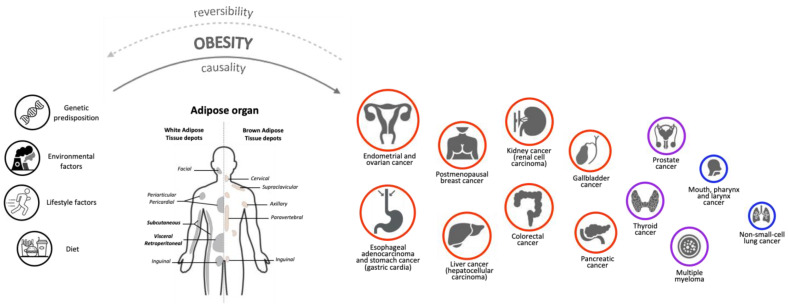
The role of Adipose Organ in the relationship between Obesity and Cancer. Experimental data indicate that, in addition to genetic, environmental, diet and lifestyle factors, the adipose organ is a key player in the relationship between obesity and cancer in humans. The 17 different cancer types we considered in the present review are represented by circles of different size and colour, based on their association with obesity. Positive associations are indicated by red circles (higher degree) or purple circles (lower degree); negatives association are indicated by blue circles. The degree of association was determined using information from the Continuous Update Project (CUP) database “https://www.aicr.org/research/the-continuous-update-project” (accessed on 24 January 2023), the National Institute of Health website “https://www.cancer.gov/about-cancer/causes-prevention/risk/obesity/obesity-fact-sheet” (accessed on 24 January 2023) and specific literature (see the main text). Given these associations, we wonder whether weight loss interventions (through bariatric surgery, pharmacotherapy, diet and/or exercise) could revert patients to the healthy condition both for obesity and cancer diseases. However, this has not been fully demonstrated, suggesting that the effects of obesity on cancer depend on multiple exogenous and endogenous factors. Further studies will be necessary to assess whether obesity has a causal effect and obesity therapy a reversible effect on cancer development.

**Figure 2 biomedicines-11-01319-f002:**
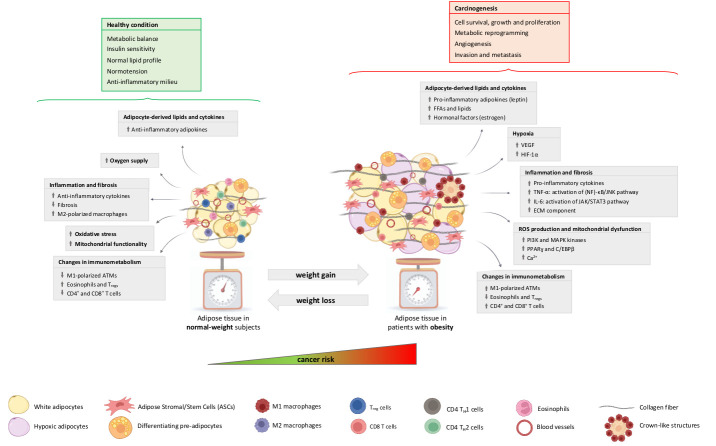
The impact of the obesity-altered adiponiche on carcinogenesis. Weight gain leads to obesity and metabolic disorders whose hallmark is adipose tissue (AT) dysfunction (adiposopathy), which may increase the risk of cancer. Main biological mechanisms of crosstalk between obesity-altered cells of the adiponiche and tumour cells are reported in grey boxes. As depicted in the figure, AT expansion and dysfunction lead to multiple effects represented by the altered secretion of adipocyte-derived lipids and cytokines—upregulation of leptin, FFAs, lipids and oestrogen; hypoxia—the upregulation of VEGF and HIF-1α; inflammation and fibrosis—increased production of pro-inflammatory cytokines, such as TNF-α (activation of (NF)-κB signalling pathway) and IL-6 (stimulation of JAK/STAT3 pathway); ROS production and mitochondrial dysfunction—the upregulation of PI3K, MAPK, PPARγ, and C/EBPβ and calcium increase; changes in immunometabolism—an increase in M1-polarised ATMs and T cells and a decrease in eosinophils and Tregs. These described mechanisms have important implications in the development of cancer, contributing to cell survival, growth and proliferation, metabolic reprogramming, angiogenesis, cell invasion and metastasis. Moreover, weight loss restores the AT phenotype to its state in normal-weight subjects, characterised by an increased oxygen supply and mitochondrial functionality, the release of anti-inflammatory cytokines, and reduced oxidative stress, inflammation and fibrosis. Considering that this reverted condition can be associated with patients’ metabolic improvement and the recovery of a healthy condition, a future aim is to assess whether weight loss can provide similar benefits across cancer patients by reverting their malignant condition. FFAs: free fatty acids; VEGF: vascular endothelial growth factor; HIF-1α: hypoxia-inducible factor 1α; TNF-α: tumour necrosis factor α; (NF)-κB: nuclear factor κB; IL-6: interleukin 6; JAK: Janus kinase; STAT3: signal transducer and activator of transcription 3; ECM: extracellular matrix; PI3K: phosphatidylinositol-3-kinase; MAPK: mitogen-activated protein kinase; PPARγ: peroxisome proliferator-activated receptor γ; C/EBPβ: CCAAT/enhancer binding protein β; ATMs: adipose tissue macrophages; Tregs: regulatory T cells.

**Figure 3 biomedicines-11-01319-f003:**
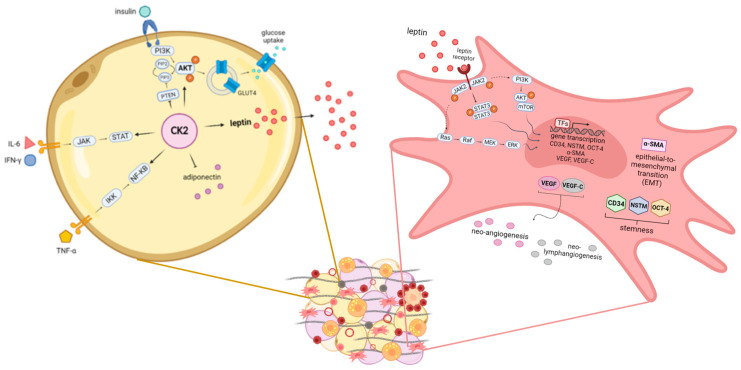
Obesity-altered adiponiche and tumour development: some novel proposed mechanisms in mature adipocytes and precursor cells. The adiponiche alterations present in patients with obesity could influence tumour biology and progression in different ways. We recently showed some pathways activated in mature adipocytes and in precursor cells that could be relevant in patients with obesity and cancer. In particular, in hypertrophic adipocytes (yellow cell on the left), CK2 potentiates insulin signalling and glucose uptake through the stimulation of PI3K/AKT pathway, GLUT4 translocation and PTEN inhibition. In addition, CK2 contributes also to IL-6, INF-γ and TNF-α signal transduction cascades, potentiating JAK/STAT and IKK/(NF)-кB signalling pathways, respectively. Moreover, CK2 is involved in regulating the production of adipokines in large adipocytes, leading to the downregulation of adiponectin and the upregulation of leptin. We showed that in the peritumoral AT of patients with EAC, leptin activates different signalling pathways in stromal stem cells (pink cell on the right). In particular, the activation of JAK2/STAT3, PI3K/AKT/mTOR and Ras/Raf/MEK/ERK leads to the increased expression of factors involved in stemness (CD34, NSTM and OCT-4), the epithelial-to-mesenchymal transition (α-SMA), neo-angiogenesis (VEGF) and neo-lymphangiogenesis (VEGF-C). CK2: casein kinase 2; PI3K: phosphatidylinositol-3-kinase; AKT: protein kinase B; GLUT4: glucose transporter type 4; PTEN: phosphatase and tensin homolog; IL-6: interleukin 6; INF-γ: interferon γ; TNF-α: tumour necrosis factor α; JAK: Janus kinase; STAT: signal transducer and activator of transcription; IKK: IκB kinase; (NF)-κB: nuclear factor κB; mTOR: mammalian target of rapamycin; Ras: rat sarcoma; Raf: rapidly accelerated fibrosarcoma; MEK: mitogen-activated protein kinase; ERK: extracellular signal-regulated kinase; CD34: cluster of differentiation 34; NSTM: nucleostemin; OCT-4: octamer-binding transcription factor 4; EMT: epithelial-to-mesenchymal transition; α-SMA: alpha-smooth muscle actin; VEGF: vascular endothelial growth factor; VEGF-C: vascular endothelial growth factor C.

**Table 1 biomedicines-11-01319-t001:** Obesity and cancer risk. Epidemiological studies showed that different sites or types of cancers displayed a higher (in red) or lower (in purple) positive or a negative (in blue) association with obesity (see Figure 1). RR: relative risk; HR: Hazard Ratio; CI: confidence index; BMI: body mass index, GERD: gastroesophageal reflux disease; M: male; F: female; MAFLD: metabolic-associated fatty liver disease; PSA: prostate-specific antigen. (Appendix B).

Cancer Site or Type	RR for Cancer Incidence (CI)	RR of Mortality or HR (CI)	Further Features	Controversial Matters with Obesity	Association with Obesity	References
Endometrial cancer	2.89 (2.62–3.18) * 1.59 (1.50–1.68) ^§^	BMI 35–39.99 Kg/m^2^: 2.77 (1.83–4.18) BMI ≥ 40 Kg/m^2^: 6.25 (3.75–10.42) °	Worse outcome		Higher degree	[27,28,39,40,41]
Ovarian cancer	1.14 (1.03–1.27) * 1.03 (0.99–1.08) ^§^	BMI 35–39.99 Kg/m^2^: 1.51 (1.12–2.02) °	Association with visceral adiposity		Higher degree	[28,42,43]
Esophageal adenocarcinoma and stomach cancer (gastric cardia)	M: 1.52 (1.33–1.74) ^§^ F: 1.51 (1.31–1.74) ^§^	Hazard Ratio 1.08 (0.77–1.52) ^	Most important risk factor: GERD; association with visceral adiposity	Better survival	Higher degree	[11,20,28,44]
Postmenopausal Breast cancer	1.40 (1.31–1.49) * 1.12 (1.08–1.16) ^§^	BMI 35–39.99 Kg/m^2^: 1.70 (1.33–2.17) BMI ≥ 40 Kg/m^2^: 2.12 (1.41–3.19) °	The inflammation combined with a higher aromatase activity increases estrogen production and insulin-resistance	Protective effect of BMI during premenopausal period	Higher degree	[28,40,45,46,47,48,49,50]
Liver cancer (Hepatocelllular carcinoma)	M: 1.24 (0.95–1.62) ^§^ F: 1.07 (0.55–2.08) ^§^	BMI 35–39.99 Kg/m^2^: M: 4.52 (2.94–6.94) F: 1.68 (0.93–3.05) °	Develop from MAFLD		Higher degree	[27,29,51,52,53,54,55]
Kidney cancer (renal cell carcinoma)	M: 1.24 (1.15–1.34) ^§^ F: 1.34 (1.25–1.43) ^§^	BMI 35–39.99 Kg/m^2^: M: 1.70 (0.99–2.92) F: 1.70 (0.94–3.05) °	M: association with increased levels of BMI, blood pressure, glucose and triglycerides F: association with BMI	Better survival	Higher degree	[28,29,56,57]
Colorectal cancer	M: 1.24 (1.20–1.28) ^§^ F: 1.09 (1.05–1.13) ^§^	BMI 35–39.99 Kg/m^2^: M: 1.84 (1.39–2.41) F: 1.36 (1.06–1.74) °		Best outcome	Higher degree	[13,27,28,29,58,59]
Gallbladder cancer	M: 1.09 (0.99–1.21) ^§^ F:1.59 (1.02–2.47) ^§^	BMI 30–34.99 kg/m^2^: M: 1.76 (1.06–2.94) F: 2.13 (1.56–2.90) °	Obesity increases the risk of gallstones		Higher degree	[27,29,60]
Pancreatic cancer	M: 1.16 (1.05–1.28)^§^ F: 1.10 (1.02–1.09)^§^	BMI 30–34.99 kg/m^2^: M: 1.49 (0.99–2.22) F: 1.41 (1.01–1.99) BMI ≥ 40 Kg/m^2^: F: 2.76 (1.74–4.36) °			Higher degree	[28,29,61]
Prostate cancer	1.03 (1.00–1.07) ^§^	BMI 35–39.99 Kg/m^2^: 1.34 (0.98–1.83) °	High BMI was positively related with non-metastatic high-grade cancer	Inverse association between BMI and PSA	Lower degree	[27,28,62,63]
Thyroid cancer	M: 1.33 (1.04–1.70)^§^ F: 1.14 (1.06–1.23) ^§^	Lack of studies	Associated with both general and visceral adiposity	Inverse association with medullary thyroid cancer	Lower degree	[28,64,65,66]
Multiple myeloma	M: 1.11 (1.05–1.18) ^§^ F: 1.11 (1.07–1.15) ^§^	BMI 30–34.99 kg/m^2^: M: 1.71 (0.93–3.14) F: 1.44 (0.91–2.28) °	Worst outcome		Lower degree	[28,30,67]
Mouth, pharynx and larynx cancer	0.81 (0.74–0.89) ^§^ 0.94 (0.57–1.56)° Adjusted for sex	HR 0.59 (0.33–1.05) ^		Data were not adjusted for smoking/drinking alcohol	Negative association	[20,68,69,70]
Non-small-cell lung cancer	Lack of studies with this specific histotype	Lung in general: HR 0.86 (0.76–0.98) ^		Positive prognostic factor, higher survival, metformin use	Negative association	[20,71,72,73,74,75,76]

* With a 10 kg/m^2^ increase in BMI; ^§^ With a 5 kg/m^2^ increase in BMI; ° RR of mortality compared with patients with a normal weight; ^ HR greater than 1 associated with a worse outcome among patients with obesity versus those without.

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
