# Peer review of "Obesity, the Adipose Organ and Cancer in Humans: Association or Causation?"

_biomedicines, 2023, doi:10.3390/biomedicines11051319_

Round 1
Reviewer 1 Report
Figure 1: This Figure is titled “Obesity and cancer: association or causation?” but no image of “obesity”. The authors may update the Figure by pathology of obesity with unhealthy adipose tissue expansion. Otherwise, the author should only refer to “Effect of adipose tissue on cancer” as an example in this Figure.
Figure 2: The text in the box are too small to read. Please enlarge the box with the text for higher resolution.
There is no table in the current manuscript. A summary table of the relationship between cancer type and obesity phenotypes for section 2 along p4-p7 would be helpful.
Figure 3: This Figure focuses on “CK2 signaling” in white adipocyte communicating with ASCs, but not on tumor development or tumor tumor-niche (tumor microenvironment). Readers will understand better if Figure 3 in section 3 along p7~p14 contains the metabolic signaling, cytokines, and adipokines in heterogeneous cell populations, including several adipocytes and immune cells that communicating with tumors.
An interesting perspective would be added in the current manuscript if some of the effects of adipose tissue on cancer cachexia were mentioned.
Author Response
Reviewer 1
Figure 1: This Figure is titled “Obesity and cancer: association or causation?” but no image of “obesity”. The authors may update the Figure by pathology of obesity with unhealthy adipose tissue expansion. Otherwise, the author should only refer to “Effect of adipose tissue on cancer” as an example in this Figure.
We thank the Reviewer for the useful comment of Figure 1. We modified Figure 1 and its title accordingly.
Figure 2: The text in the box are too small to read. Please enlarge the box with the text for higher resolution.
We edited Figure 2 following Reviewer’s suggestions.
There is no table in the current manuscript. A summary table of the relationship between cancer type and obesity phenotypes for section 2 along p4-p7 would be helpful.
In agreement with the Reviewer’s comment, we added a specific Table summarizing epidemiological data and association’s strength between obesity condition and cancer subtypes (Table 1). We think that this new Table 1 could completely substitute the text from 2.1 to 2.13 (lines 164-298), in order to shorten and streamline the Review, if the Reviewers and the Editor agree. The text could be added as on line supplementary material.
Figure 3: This Figure focuses on “CK2 signaling” in white adipocyte communicating with ASCs, but not on tumor development or tumor tumor-niche (tumor microenvironment). Readers will understand better if Figure 3 in section 3 along p7~p14 contains the metabolic signaling, cytokines, and adipokines in heterogeneous cell populations, including several adipocytes and immune cells that communicating with tumors.
We thank the Reviewer for the comment that allow us to better explain the role of Figure 3 in the Review context. In Figure 2 we depicted the adipose tissue niche with the main cell types contained inside (mature adipocytes, adipose stromal/stem cells, immune cells) and matrix/environmental elements underlining their possible roles, with different mechanisms, in carcinogenesis (text boxes on the right). In Figure 3 we would like to highlight some novel specific molecular mechanisms emerging from the studies conducted by our group that could have a role on tumour development. In particular, we studied the CK2 signalling in mature adipocytes that could modify the functional features and the secretome of these cells acting on different pathways in which this pleiotropic kinase is involved. On the other hand we showed the activation of specific pathways leading to the increasing the expression of stemness, neo-angiogenesis and lymphangiogenesis factors which in turn could act on tumour cells. Thus, we showed these alterations in cells of the adiponiche of patients with obesity (as depicted in Figure 3 at the bottom) and we hypothesized that they could play a role on tumour biology in patients with obesity and cancer.
In order to better clarify the mechanisms depicted in Figure 3 we eliminated the connection via leptin suggesting an adipo-niche internal cross talk between CK2 signalling and leptin activation of specific pathways in adipose tissue stromal/stem cells. Moreover, we changed accordingly the Figure legend.
An interesting perspective would be added in the current manuscript if some of the effects of adipose tissue on cancer cachexia were mentioned.
According to the Reviewer’s suggestion, we added a paragraph concerning the effects of the adipose organ on cancer cachexia (new paragraph 6, lines 548-573). This very interesting topic lead us to underline and discuss another aspect of the controversial role of obesity in patient with cancer (the obesity paradox) taking into account the “beneficial effect” and the “detrimental role” of the increased fat mass and sarcopenia.
Reviewer 2 Report
The present manuscript entitled “Obesity, adipose organ and cancer in humans: association or 2 causation?” that outlines an important contribution to the literature in the field of obesity and cancer. There have been many research papers on the obesity in the progression of cancer. Therefore, to improve the quality of this article author should discuss the molecular mechanism of how obesity is the causative factors for cancer? Authors must emphasize the new findings and novelties of this manuscript.
All the figures are looking good. Overall, the paper is technically sound, thoughtful and generally supports their conclusions. After, addressing the comments the current manuscript has significance for the publication in this esteemed journal.
Author Response
Reviewer 2
The present manuscript entitled “Obesity, adipose organ and cancer in humans: association or causation?” that outlines an important contribution to the literature in the field of obesity and cancer. There have been many research papers on the obesity in the progression of cancer. Therefore, to improve the quality of this article author should discuss the molecular mechanism of how obesity is the causative factors for cancer? Authors must emphasize the new findings and novelties of this manuscript.
All the figures are looking good. Overall, the paper is technically sound, thoughtful and generally supports their conclusions. After, addressing the comments the current manuscript has significance for the publication in this esteemed journal.
We thank the Reviewer for underlining the interest and the utility of our manuscript. We tried to highlight the epidemiological and clinical data supporting the causative role of obesity in cancer development, but refer also the controversial evidence indicating a protective effect of obesity especially for specific cancer subtypes. Conclusive results are not available now and studies attesting a relevant cancer remission or a reduced cancer risk in patients underwent to weight loss and body mass index (BMI) normalization due to obesity therapy are lacking or incomplete.
We focused our attention on the role of adipose organ and in particular of adipose tissue niche in the development of tumour considering the main cellular processes and molecular mechanisms involved (described in the paragraph 3, lines 185-480). Moreover, we collected evidence regarding other systemic alterations (microbiota and circadian rhythms dysregulation) recently investigated in obesity and their possible involvement in tumour biology and cancer progression (described in paragraphs 4 and 5, respectively; lines 481-512 and 513-547). As requested by the Reviewer, we emphasized the new findings and novelty in the conclusion of the manuscript (paragraph 8, lines 648-657).
Reviewer 3 Report
It is quite an interesting manuscript. The topic of this manuscript falls within the scope of Biomedicines
The appropriate tables and figures have been provided. The article is easy to read and logically structured. The conclusions are consistent with the presented evidence and arguments. The figures included are clear and easy to understand;
There are only some comments in the reviewer's opinion that should be taken under consideration by the Author:
1.what are the limitations of the studies included in this review?
2. How does obesity affect cancer survivors?
3. future directions-please cite the paper (https://www.mdpi.com/2072-6694/15/2/485
Author Response
Reviewer 3
It is quite an interesting manuscript. The topic of this manuscript falls within the scope of Biomedicines. The appropriate tables and figures have been provided. The article is easy to read and logically structured. The conclusions are consistent with the presented evidence and arguments. The figures included are clear and easy to understand.
There are only some comments in the reviewer's opinion that should be taken under consideration by the Author:
1.what are the limitations of the studies included in this review?
- How does obesity affect cancer survivors?
- future directions-please cite the paper (https://www.mdpi.com/2072-6694/15/2/485)
We thank the Reviewer for underlining the interest of our Review article and its appropriateness. According to the Reviewer’s suggestion, we addressed the comments adding the limitation of the studies included and inserted the reference suggested in the concluding paragraph of the manuscript (paragraph 8, lines 667-671 and 701). Lastly, we discussed how the obesity condition affects cancer survivors in the introduction section (paragraph 1, lines 121-129).
Round 2
Reviewer 1 Report
I recommend publication as a valuable review paper in Biomedicines after English proofreading.
Reviewer 3 Report
The paper may be published in its current form. The Authors included all remarks.